# Lossless Encoding of Time-Aggregated Neuromorphic Vision Sensor Data Based on Point-Cloud Compression

**DOI:** 10.3390/s24051382

**Published:** 2024-02-21

**Authors:** Jayasingam Adhuran, Nabeel Khan, Maria G. Martini

**Affiliations:** 1Faculty of Engineering, Computing, and the Environment, Kingston University London, Penrhyn Rd., Kingston upon Thames KT1 2EE, UK; j.adhuran@kingston.ac.uk (J.A.); m.martini@kingston.ac.uk (M.G.M.); 2Department of Computer Science, University of Chester, Parkgate Road, Chester CH1 4BJ, UK

**Keywords:** neuromorphic vision sensor (NVS), neuromorphic spike events, point-cloud compression, silicon retinas, spike encoding

## Abstract

Neuromorphic Vision Sensors (NVSs) are emerging sensors that acquire visual information asynchronously when changes occur in the scene. Their advantages versus synchronous capturing (frame-based video) include a low power consumption, a high dynamic range, an extremely high temporal resolution, and lower data rates. Although the acquisition strategy already results in much lower data rates than conventional video, NVS data can be further compressed. For this purpose, we recently proposed Time Aggregation-based Lossless Video Encoding for Neuromorphic Vision Sensor Data (TALVEN), consisting in the time aggregation of NVS events in the form of pixel-based event histograms, arrangement of the data in a specific format, and lossless compression inspired by video encoding. In this paper, we still leverage time aggregation but, rather than performing encoding inspired by frame-based video coding, we encode an appropriate representation of the time-aggregated data via point-cloud compression (similar to another one of our previous works, where time aggregation was not used). The proposed strategy, Time-Aggregated Lossless Encoding of Events based on Point-Cloud Compression (TALEN-PCC), outperforms the originally proposed TALVEN encoding strategy for the content in the considered dataset. The gain in terms of the compression ratio is the highest for low-event rate and low-complexity scenes, whereas the improvement is minimal for high-complexity and high-event rate scenes. According to experiments on outdoor and indoor spike event data, TALEN-PCC achieves higher compression gains for time aggregation intervals of more than 5 ms. However, the compression gains are lower when compared to state-of-the-art approaches for time aggregation intervals of less than 5 ms.

## 1. Introduction

A Neuromorphic Vision Sensor (NVS) [1] is a device imitating biological visual sensing, i.e., reporting only light intensity changes in the observed scene. Differently from conventional cameras, where frames are acquired at uniform intervals, NVSs asynchronously acquire brightness changes per pixel with microsecond resolution. Spike events are triggered in response to logarithmic illumination changes, i.e., whenever there is motion of the vision sensor, movement in the scene, or a change of light conditions in the scene. These unique properties enable NVSs to achieve an ultra-low response latency, a high dynamic range, low informative data rates, and low power consumption.

Emerging applications of NVSs can be found in diverse scenarios, ranging from autonomous cars [2] to robotics [3] and Unmanned Aerial Vehicles (UAVs) [4]. Furthermore, these sensors have the potential to replace conventional vision sensors [5] in diverse and unique computer vision applications such as smart agriculture.

Even if the biologically inspired vision-sensing technique provides an inherent compression, further reduction of data can be beneficial for transmission in Internet of Intelligent Vehicles (IoV) such as Internet of Drones (IoD), and Industrial Internet of Things (IIoT) scenarios. The rate at which spike events are triggered by these sensors depends on the scene complexity and on the sensor speed, as studied in [6,7], where a model for spike event rate estimation is also presented. The Address Event Representation (AER) protocol is used for representing and exchanging uncompressed spike data, with each event represented by a tuple (x,y,p,t), where *x* and *y* are the coordinates of the pixel where a brightness change occurred, *t* is the timestamp expressed in μs, and *p* is the polarity of the event (an increase or a decrease in light intensity). An example of event data representation as a tuple is reported in Table 1. Each tuple is represented by 64 bits (4 bytes for timestamp and 4 bytes for the remaining three fields).

To introduce the NVS data representation and show how it can be interpreted as a point cloud, Figure 1 shows an example of two different scenes of different complexity [6], and hence different point cloud density—i.e., *Shapes* and *Dynamic* from the Dynamic and Active-pixel Vision Sensor (DAVIS) dataset [8]. In the *Shapes* scene, spike events are produced when the camera is rotated in front of static 2D shapes. In the *Dynamic* scene, spike events are produced by both the motion within the scene and the rotation of the sensor. Motion and camera information of the aforementioned sequences are further elaborated in [8]. The second column of Figure 1 shows the spatial (*x*,*y*) coordinate plots of the spike events from each scene, i.e.,  *Dynamic* (above) and *Shapes* (below). Polarity zero-spike events are projected onto the 2D plot for the *Shapes* scene, whereas polarity one-spike events are reported in the *Dynamic* 2D plot. This representation is obtained by visualizing the 3D point cloud representation of the data on the (*x*,*y*) plane. The last column shows the spatio-temporal 3D plot (x,y,t), i.e., the 3D plot of the spatial and temporal coordinates of the *Dynamic* (above) and *Shapes* (below) scenes, from a perspective that enables appreciating the 3D nature of the data. The RGB images in Figure 1 do not refer to the same time instant as the aggregated point clouds, as they are representative images of the sequence provided in the DAVIS dataset.

### 1.1. Motivation

A growing number of diverse applications [2,3,9,10,11,12,13,14,15] consider the accumulation of spike events over a fixed time interval. The accumulation of spike events allows the use of state-of-the-art algorithms for applications ranging from classification to object detection and tactile sensing. Table 2 shows the applications of spike event aggregation using different algorithms on diverse tasks. For instance, digit classification [9] utilizes the SKIM algorithm, where spike events are accumulated over a period of 20 ms. On the other hand, the authors in [10] utilize a deep residual network algorithm, where spike events are aggregated over a 50 ms time interval for motion estimation in autonomous driving. These algorithms project the accumulated stream of spike events at uniform intervals in temporal frames, which are then fed to different machine learning and deep learning algorithms.

Besides allowing the use of state-of-the-art machine and deep learning algorithms, the accumulation of spike events has other advantages, i.e., it also performs inherent compression of data, which can be useful for the storage and transmission of spike events. Figure 2 shows a 3D point cloud plot (spatial and temporal coordinates of spike events with a polarity of zero) with and without spike event accumulation over 20 ms. As shown in Figure 2b, with the time aggregation of spike events, the point cloud has fewer 3D points (a less dense point cloud) as compared to the case when no accumulation is applied, as shown by the denser point cloud in Figure 2a.

Although an increasing number of applications are considering spike event accumulation, as mentioned above, most of the compression strategies consider raw spike events. As in our earlier TALVEN approach [17], we aim to fill this research gap by considering spike event accumulation over a fixed time interval, followed in this case by point-cloud compression. We proposed already to use point-cloud compression for NVS data in [18], and we propose here to extend the approach by using the time-aggregation of event data prior to point cloud compression.

### 1.2. Contribution

Our recent work, TALVEN [17], utilized spike event aggregation and video encoding, i.e., the accumulated spike event stream is projected as an event frame for each polarity. The two event frames for each polarity are concatenated as a single *superframe*. Each pixel value in a *superframe* represents the aggregated spike event count. In other words, the sequence of spike events within the time aggregation interval is represented as a *superframe*. A *superframe* can be interpreted as a 2D histogram of the event data in the aggregation interval. Since in TALVEN the spike event stream is transformed into a format mimicking a sequence of video frames, video encoding (lossless mode) is applied to the frames carrying an accumulated spike event count. TALVEN achieved superior compression gains as compared to the state-of-the-art compression approaches for aggregation intervals that are longer than a content-dependent threshold value. This is due to the efficient exploitation of spatial and temporal redundancy.

In this work, we represent the accumulated spike event stream as a point cloud with spike events for each polarity as points in an (x,y,t) 3D space. The spatio-temporal point cloud representation of neuromorphic vision sensor data is input into a point cloud encoder (see also [18]). We first reduce in size the point cloud by spike event accumulation. Further compression is applied to the accumulated data by employing a point cloud encoder. We propose to use the standardized ISO/IEC MPEG Geometry-based Point Cloud Coding (G-PCC) method on the accumulated data. This is the output of a standardization process and the relevant source code is stable and reliable as such. The benefit of utilizing a standardized encoding method would be that we expect that software and hardware encoders will be largely available in media devices, hence making then stable, reliable, and not requiring an additional cost.

The contributions provided in this work include: a novel strategy (TALEN-PCC) to losslessly compress time-aggregated NVS data based on point-cloud compression; an analysis of its performance for different time-aggregation intervals Δt; and the comparison in terms of compression gains of the proposed strategy with the state-of-the-art strategy with time aggregation [17] on outdoor and indoor scenes from a public dataset [8].

The remainder of this paper is structured as follows. Related work is discussed in Section 2, where first the state-of-the-art lossless compression strategies are briefly reviewed and an introduction to point-cloud compression follows. Our compression approach is proposed in Section 3. Section 4 reports the evaluation setup and the considered datasets used to assess the proposed and benchmark compression algorithms. The compression gain of our strategy and benchmark are presented and discussed in Section 5. Finally, Section 6 concludes the paper.

## 2. Related Work

### 2.1. NVS Data Compression

The authors in [19] proposed the first compression method specifically designed for neuromorphic vision sensor data, called Spike Coding. The Spike Coding algorithm is based on the spike firing model of the neuromorphic vision sensor. However, the encoding of spike events via this method achieves limited compression gains, i.e., compression ratios in the range of 2 to 3 are achieved on the dataset of intelligent driving by the Spike Coding method [20].

In a previous work [17], we proposed to organize spike events in a sequence of frames. We proposed to aggregate spike events into frames, in such a way that each pixel value in a frame represents the event count. For each polarity, the combination of frames over time can be seen as a video sequence. After combining the obtained frames into *superframes*, video coding is applied to the aggregated spike event stream. The combination of time aggregation and video encoding exploits temporal and spatial correlation within the sequence of spike event stream, thus resulting in superior compression gains as compared to the state-of-the-art compression methods.

A similar event aggregation strategy was used in [21], but  adding events in each (*x*,*y*) position with their signed polarity, and hence reducing the information in each (*x,y*) position to three possibilities (no event, positive polarity, or negative polarity). They used small aggregation intervals (results are provided for frame rates corresponding to 1 to 5 ms). A different compression method was then used on this representation. The compression ratios appear to have been calculated based on raw data in the frames rather than the original event raw data. The same authors proposed a low-complexity method  [22] with a performance that was slightly better than Lempel–Ziv–Markov chain algorithm (LZMA) (that we have seen underperforms vs. our TALVEN approach, used here as a benchmark). A similar strategy was used in [23].

Other compression methodologies can be tailored to NVS data. These include advanced dictionary coders, such as Zstandard (Zstd) [24], Zlib [25], LZMA [26], and Brotli [27]. The output of the NVS is a multivariate stream of integers; therefore, integer-compression algorithms such as Simple8B [28], Memcpy [28], Single-Instruction Multiple Data (SIMD)-based strategies (such as SIMD-BP128 [29], and FastPFOR [29]), and Snappy [30] can be applied to the spike event stream. IoT-specific compression strategies, such as Sprintz in [28], can also be tailored to the neuromorphic spike event stream. Compression performance results for NVS data of these approaches have been provided in [31].

In [18], we recognized the possibility of treating NVS data as a spatio-temporal point cloud, which can be compressed via methods developed for volumetric point clouds. While we considered there different point cloud sizes (i.e., splitting a large point cloud into smaller point clouds in the time domain), we did not consider time aggregation and focused on fully lossless compression. The same approach was very recently used in [32], where a different point cloud codec (Draco) was also tested in addition to G-PCC, and the results were compared with the benchmark results we provided in [31].

### 2.2. Point Cloud Compression

A 3D point cloud is a set of points in the 3D space, each represented by the coordinates (x,y,z), with possibly attributed information (e.g., representing color). In the case of dynamic content, a different point cloud is considered at each time instance kΔt, representing the time variation of a point cloud.

Point clouds can have an extremely large number of points, resulting in huge file sizes and data rates and hence costs for storage and transmission.

Different methods exist to compress point cloud data [33,34,35]; most of them perform compression of point cloud geometry using octree coding [36], where data are transformed into voxel representation, in order to appropriately exploit volumetric redundancy, and are then partitioned until sub-cubes of dimension one are reached; local approximations called “triangle soups” (trisoup) can be adopted, where the geometry can be represented by a pruned octree plus a surface model [36,37].

When lossy compression is adopted, distortion metrics can be obtained for instance from the symmetrized point-to-point or point-to-distance mean squared error, which is converted to PSNR using the original point cloud bit depth as the peak error [38].

The Moving Picture Expert Group (MPEG) standardized a point-cloud encoder assuming that data are represented as coordinates in a 3D space (x,y,z), plus reflectance and RGB attributes for each point. Two main proposals were developed [39]: Video-based, which is appropriate for point sets with a relatively uniform distribution of points (V-PCC); and geometry-based (G-PCC) [40], which is appropriate for more sparse distributions. G-PCC decomposes the 3D space into a hierarchical structure of cubes, and each point is encoded as an index of the cube it belongs to; it has the advantage of a native 3D representation.

Detailed reviews and an analysis of point-cloud compression approaches, in particular in MPEG, can be found in [35,39,41], and a good overview of the standardization activities is provided in [42].

## 3. Proposed Strategy

The proposed method of compression is shown in Figure 3. We propose here to (1) split the NVS stream into two streams, each associated with a different polarity, (2) select an aggregation time interval Δt, and (3) obtain for each polarity a representation of the NVS data as points in spatio-temporal 3D space positions [x,y,kΔt], where each of these positions is characterized by the number of events in position (x,y) in the interval of duration Δt between timepoints (k−1)Δt and kΔt. We have at this point two point clouds, one per polarity, composed of time-aggregated NVS data reported in a space–time (x,y,kΔt) tridimensional space as a point cloud. (4) We perform on each of them lossless compression based on point-cloud compression. In this case we propose to use the G-PCC strategy  [40], using the attribute field (16 bits) associated with each point in the cloud to represent the number of events aggregated. The main steps of the proposed strategy are discussed in the following subsections.

### 3.1. Spike Event Aggregation

The raw spike event data, [x,y,p,t], undergo aggregation, i.e., all the spike events within the time-aggregation interval of Δt are accumulated. The accumulation is performed by associating the event count with the respective spatial position (x,y). For instance, consider the sample event stream shown in Table 1. After applying accumulation, the spatial locations of (7, 17), (1, 10), and (2, 20) of the polarity zero matrix will have event counts of 1, 2, and 1, respectively. Similarly, the spatial positions of (5, 18), (8, 20), and (8, 11) of the polarity one matrix will have an event count of one.

### 3.2. Multivariate Stream

The next step is the creation of a multivariate stream from polarity one and zero matrices. We apply a raster scan to the matrices in such a way that spatial locations with a zero event count are filtered, i.e., only non-zero event count locations are converted into a multivariate stream. Spatial locations are converted into a multivariate stream with four variables, [x,y,Eventcount,kΔt].

Figure 3 shows the creation of a multivariate stream. After the raster scan (Flag 0 matrix), the spatial location (1, 7) has an event count of 1, followed by (1, 8) with an event count of one. In the second row, the spatial location (2, 2) has the non-zero event count of 3. Figure 3 shows the multi-variant stream with a time-aggregation interval of 1 ms (Δt=1). According to the figure, the raw spike event stream [x,y,p,t] is transformed into two aggregated streams, [x,y,Eventcount,kΔt], of flags zero and one (shown in Figure 3).

### 3.3. G-PCC Encoding

The next step is to apply compression to the two streams. The aggregated multivariate stream can be compressed using any multivariate compression algorithm such as dictionary-based compression (LZMA or Zlib) or fast integer compression. In other words, the spike event-aggregation and multivariate stream-extraction steps can be combined with any dictionary or integer compression algorithm. We propose to apply a geometry-based 3D point cloud encoder. Our previous work in [18] shows excellent performance of the geometry-based point-cloud encoder (G-PCC) on the raw spike event stream. G-PCC encoding outperforms LZMA and spike coding. The G-PCC compression strategy has two inputs, namely position and attribute. The position input takes the 3D, (x,y,z), volumetric data, whereas the attribute input takes color or reflectance input associated with the 3D points. G-PCC supports lossless compression of the volumetric data; therefore, we utilize the lossless mode of the encoder.

Rather than using the encoder on spatial 3D data, we propose to apply spatial and temporal coordinates, [x,y,kΔt], to the position input of the G-PCC encoder. In other words, we utilize the G-PCC encoder while using the z-axis for the evenly spaced time-aggregation instants. The G-PCC encoder exploits the volumetric redundancy within the 3D point cloud stream. The event count is fed into the reflectance input of the G-PCC encoder, as shown in Figure 3. We observe that we can represent a maximum value of 65536 events at each spatial location (per time aggregation interval) with the 16 bit attribute field.

The first step of the volumetric data processing is the coordinate transformation. After the coordinate transformation of the 3D points, voxelization and geometry analysis steps are performed as shown in Figure 4. The encoder utilizes the concept of octree decomposition, where 3D point clouds are decomposed recursively into subcubes. The encoder finds the best-matching subcubes, i.e., the correlation between 3D point shapes is exploited by finding the subcubes that have a similar geometry. The final step of the G-PCC encoder is the entropy coding, where the decomposed subcubes are fed into the coder to further exploit the redundancies. The resultant stream of the G-PCC encoder is composed of a geometry bitstream and an attribute bitstream, as shown in Figure 4.

**Figure 4 sensors-24-01382-f004:**
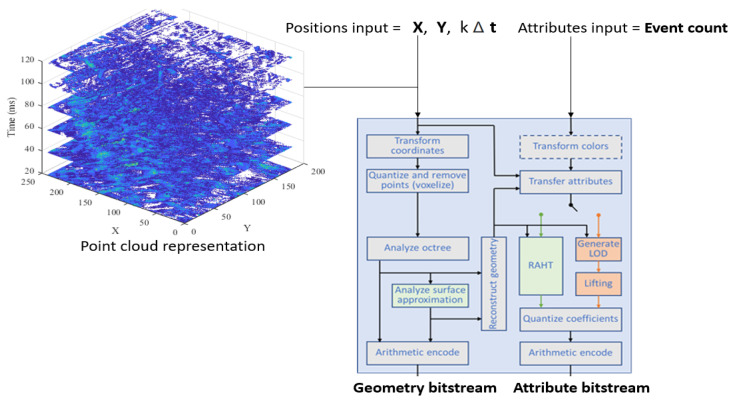
Positions and attributes input into the G-PCC point-cloud encoder [40], where Δt=20 ms, $k_{max}=6$, and $T_{seq}=k_{max}\times \Delta =120$ ms.

### 3.4. Implementation Details

The summary of all the steps of the aggregation-based point cloud encoding is shown in Algorithm 1. The aggregation time interval, Δt, depends for instance on the machine or deep learning algorithm to be used for a particular application. The total number of iterations, $k_{max}$, depends on the spike event sequence duration $T_{seq}$ and the accumulation time interval Δt. In each iteration (for loop), the spike events are accumulated over the time interval Δt. After each iteration, a raster scan is applied to produce the multivariate stream of spatio-temporal coordinates and spike event count. After all the iterations (end of for loop), the resultant multivariate stream is fed into the G-PCC encoder, as shown in Algorithm 1. The graphical illustration of Algorithm 1 is shown in Figure 4, where Δt=20 ms, $k_{max}=6$, and $T_{seq}=k_{max}\times \Delta =120$ ms. The point-cloud representation of the positions input is shown in Figure 4. The G-PCC encoder exploits the volumetric redundancies within the discrete set of spatiotemporal points.

### 3.5. Computation Complexity

The computation complexity of the proposed TALEN-PCC strategy is a function of the three processing steps discussed in Section 3.1, Section 3.2 and Section 3.3. The complexity in the spike event aggregation step (Section 3.1) is a linear function of the total number of input events that are projected in polarity one and zero matrices. In the creation of the multivariate stream (Section 3.2), a raster scan is applied to the matrices; therefore, the processing complexity is dependent on the resolution of the matrix. The complexity of this step is twice the spatial resolution, i.e., 2×M×N. The complexity of the final step of TALEN-PCC is dependent on the G-PCC encoder. According to [41], the lossless compression mode of 3D geometric compression (lossless geometry) requires minimal computation with lower time complexity. Moreover, the hardware acceleration and parallel optimization of the proposed TALEN-PCC method can further reduce the complexity and speed challenges. It is important to note that the computation complexity of TALVEN is comparatively higher than TALEN-PCC, mainly because of the higher complexity of the motion compensation of the *superframes*.
**Algorithm 1** TALEN-PCC**Input:** Spike event data stream with $N_{events}$  **Input:** Aggregation time interval Δt [ms]  **Input:** DVS spike sequence duration $T_{seq}$ [ms]  **Input:** $k_{max}=\frac{T_{seq}}{\Delta t}$  **for** k=1 to $k_{max}$ **do**   **while** t≤kΔt **do**       Accumulate flag 0 events. Write event count in an M×N matrix for flag 0       Accumulate flag 1 events. Write event count in an M×N matrix for flag 1     **end while**    Apply raster scan to flag 0 matrix. Extract spatial coordinates (x, y), and event count from the raster scan of flag 0 matrix. Multivariate stream of x, y, Event-count, kΔt for flag 0 is input to a file.     Apply raster scan to flag 1 matrix. Extract spatial coordinates (x, y), and event count from the raster scan of flag 1 matrix. Multivariate stream of x, y, Event-count, kΔt for flag 1 is input to a file.  **end for** Multivariate streams extracted from the raster scans of both the matrices are fed to GPCC encoder.  Spatial coordinates (x, y), and kΔt are fed to the positions input of the GPCC encoder.  Event count is fed to the Attributes input of the GPCC encoder.  **Output:** Polarity 1 compressed bitstream, with size $\gamma_{1}$ (bits)  **Output:** Polarity 0 compressed bitstream, with size $\gamma_{0}$ (bits)  **Output:** Total size (bits) of the output stream, $\gamma =\gamma_{0}+\gamma_{1}$

## 4. Performance Evaluation Setup

### 4.1. Dataset

Comparative compression performance analysis of the proposed and benchmark strategies was conducted on the Dynamic and Active-pixel Vision Sensor (DAVIS) dataset [8]. The dataset has diverse indoor and outdoor scenes with varied sensor motion, ranging from rotational to translational, etc. We extracted sequences with diverse scene complexities and sensor motion speeds. The proposed and benchmark compression algorithms were applied to the extracted sequences reported in Table 3.

### 4.2. Data Processing

The AER data format is utilized by the DAVIS sensor for the representation of the spike event stream. According to the AER format, each spike event is 64 bits long. Temporal information is represented by 32 bits, whereas spatial information is represented by 19 bits, i.e., *x* and *y* spatial information is represented by 10 and 9 bits, respectively. The polarity flag and polarity change (the polarity change is set to one when the polarity flag switches from zero to one or one to zero) information is represented by 2 bits. The remaining 11 bits are used to represent sensor information such as temperature, gyroscope, and acceleration. Further details related to the 8 byte spike event are reported in [31].

Data acquired by the sensors was converted from a series of 64 bit AER data into a multivariate stream (x,y,p,t) of spike events, which we used as input for our method. Figure 6 in [31] illustrates AER vs. four-tuple conversion.

### 4.3. Benchmark Strategies

We have observed in [17] that the TALVEN compression approach outperforms the other lossless compression strategies. For this reason, and since TALVEN is the only other compression strategy based on time aggregation, we compare the proposed strategy with the TALVEN strategy. We also compare the proposed TALEN-PCC strategy with the Spike Coding (SC) algorithm [19]. SC is the the first compression algorithm designed specifically for the spike event stream from neuromorphic vision sensors. The proposed and the benchmark strategies were evaluated at six different time-aggregation intervals of 1 ms, 5 ms, 10 ms, 20 ms, 40 ms, and 50 ms.

### 4.4. Compression Ratio

The performance of the considered and benchmark strategies was evaluated by computing the end-to-end compression ratio CR. For a scene of duration $T_{seq}$,
(1)$$CR=\frac{N_{events}\times 64}{\gamma }=\frac{N_{events}\times 64}{\gamma_{0}+\gamma_{1}}$$
where γ is the total size (in bits) of the compressed output stream of the time duration *T*, $\gamma_{0}$ and $\gamma_{1}$ represent the size (in bits) of the compressed output streams associated with polarities 0 and 1, respectively, and $N_{events}$ is the total number of spike events in the time duration *T*, with each event requiring 64 bits for representation in uncompressed mode.

## 5. Results

The compression performance analysis is divided into two parts. In the first part, we analyze the compression performance with respect to the scene complexity and event rate at a fixed time aggregation interval. Section 5.1 reports the comparative compression performance of different outdoor and indoor scenes with varied scene complexities and spike event rates. In the second part (Section 5.2), we compare the compression performance of the proposed strategy with the benchmark strategies at different time-aggregation intervals of 1 ms, 5 ms, 10 ms, 20 ms, 40 ms, and 50 ms.

### 5.1. Compression Gain Analysis at Δt=20ms

Figure 5 shows the end-to-end compression performance for the ten scenes in Table 3, when neuromorphic events are aggregated over a time interval of 20 ms. The figure reports via a radar plot the comparison of compression ratios (reported in the radii) for the proposed point cloud-based strategy and our previously proposed TALVEN strategy (based on the principles of video compression). For all sequences, the proposed point cloud-based strategy outperformed the state-of-the art TALVEN strategy, in particular for scenes with a low event rate, such as *Shapes*. As observed in previous works, the compression ratios were highly variable with the content of the scenes.

The *Boxes* and the *Poster* scenes yielded the highest compression gains owing to the highest event rate of more than 4 mega-events per second among all the considered scenes. A high event rate results in a higher accumulation of spike events, which in turn results in a higher event count per spatial location, thus reducing the accumulated stream. Therefore, both strategies achieved the highest compression ratio for the *Boxes* and *Poster* scenes, as shown in Figure 5. Compression performance was highly dependent on the scene complexity. For instance, the *Shapes* scene has the lowest scene complexity along with low motion of the sensors, which resulted in the lowest event rate of 246.61 Kev/s among the considered scenes. The *Shapes* scene resulted in a higher compression gain of 29.21 for TALVEN and 34.34 for TALEN-PCC. Both strategies exploit the spatial and temporal correlation of the low-complexity *Shapes* scene, thus resulting in one of the highest compression gains among the considered scenes. The *Dynamic* scene has a higher event rate (4.4 times higher than that of the *Shapes* scene); however, the compression gain is approximately similar to that of the *Shapes* scene. Intuitively, a higher event rate should yield a higher compression gain. However, the higher scene complexity of the *Dynamic* scene limits the compression gain. Slider yielded the lowest compression gain among the considered indoor scenes, owing to a lower event rate (low speed of the sensor) and higher scene complexity. The outdoor *Urban* and *Walking* scenes have approximately the same sensor motion. However, the scene complexity of the *Urban* scene is more dense, which results in higher event rate for the *Urban* scene as compared to the *Walking* scene. Therefore, both strategies resulted in a higher compression gain for the *Walking* scene, as shown in Figure 5, mainly because of its low scene complexity. The scene complexity of all three *running* scenes is approximately the same; therefore, the compression performance depends on the event rate. The *Running1* scene has the lowest sensor motion (lowest event rate) among the three scenes, whereas the *Running3* scene has the highest speed. Both strategies resulted in a higher compression gain for the *Running3* scene followed by *Running2* and *Running1*, as shown in Figure 5.

### 5.2. Comparative Performance Analysis of the Proposed and Benchmark Strategies

The compression performance of the proposed and benchmark strategies at different time accumulation intervals is shown in Figure 6. The spike-coding strategy yielded better compression gains at lower time aggregation intervals. This is mainly because the SC strategy utilizes macro-cubes (also called event frames), where a fixed number of spike events is projected. Spike event traffic from neuromorphic vision sensors is highly variable, i.e., the traffic is bursty in nature. The projection of a fixed number of events (32,768 spike events as studied in [20]) per macro-cube results in a fixed number of event frames irrespective of the time-accumulation interval. For instance, a 1 ms accumulation interval results in 1000 frames per second for TALVEN. On the other hand, 32,768 events per macro-cube resulted in only 7.5 event frames per second if we consider the *Shapes* sequence. The higher the number of frames, the higher the overhead information, which results in limited compression gains (at lower aggregation intervals) for the TALVEN strategy. SC yields limited compression gains at higher time-aggregation intervals. This is mainly because the SC strategy does not take advantage of spike event accumulation, which results in limited compression gains at higher time intervals, as shown in Figure 6.

According to Figure 6, the increase in the time aggregation interval size increases the compression gains for TALVEN and TALEN-PCC. This is mainly because as the aggregation interval increases, more spike events accumulate, which boosts the compression gain. At lower time-aggregation intervals, for instance 1 ms, the performance difference between TALEN-PCC and TALVEN is the highest. The number of frames per second is very high for TALVEN, i.e., 1000 fps for 1 ms. These frames carry spike event count information at each spatial location. The low-event rate sequences (*Shapes*, *Slider*, *Urban*, and *Walking*) resulted in a higher proportion of empty spatial locations (zero event count) within a frame, which in turn resulted in limited compression gains for TALVEN. TALEN-PCC resolved this issue by filtering out zero-event count spatial locations by applying the raster scan, thus resulting in better compression gains for lower- and medium-event rate sequences, as shown in Figure 6. TALVEN also suffers from higher overhead information related to each frame. Furthermore, TALVEN exploits inter-frame correlation between the neighboring frames, whereas TALEN-PCC exploits temporal correlation within the entire point cloud, thus resulting in better exploitation of spatial and temporal redundancies.

TALEN-PCC outperformed TALVEN for all of the considered 10 scenes. The difference in compression performance was the highest for the low-event rate sequences. On the other hand, the compression difference was minimal for the high-event rate sequences of *Poster* and *Boxes* (high-complexity scenes). This is mainly because a high scene complexity results in limited spatial and temporal correlation, and thus the difference in compression performance between TALEN-PCC and TALVEN is minimal. Both strategies resulted in higher compression gains (*Poster* and *Boxes* sequences), mainly because of the higher accumulated spike event count.

## 6. Conclusions and Future Work

In this paper, we proposed a strategy to losslessly compress, via point-cloud compression, time-aggregated spike-event data generated from NVSs. According to the experimental analysis, the proposed strategy, TALEN-PCC, shows improved compression ratios vs. the benchmark strategies of TALVEN and SC. In particular, an improved compression ratio for time aggregation-based TALEN-PCC vs. TALVEN of up to 30% was observed for the analyzed sequences. The lower were the scene complexity and event rate, the higher was the compression difference between the proposed and benchmark strategies. On the other hand, the compression difference was minimal, between TALVEN and TALEN-PCC, for high-event rate sequences. In future work, we will address and discuss aspects such as the impact of different aggregation time intervals on the compression ratio, complexity, and delay.

Future work will include an analysis of the relationship between content complexity and compression efficiency. In a previous paper [6], we studied a method to characterize the content of a scene that is linked with its data rate, and we developed a relevant model. We plan to extend the model to the case of the data rate of compressed NVS data.

## Figures and Tables

**Figure 1 sensors-24-01382-f001:**
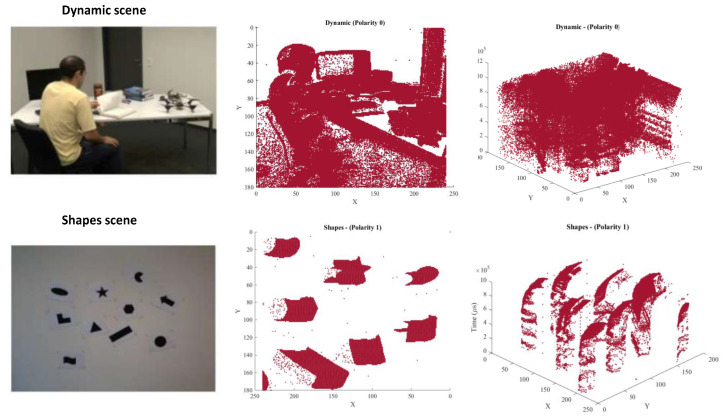
First column: *Dynamic* and *Shapes* scenes. Second column: 2D plot (x,y) of spike events from the *Shapes* and *Dynamic* scenes. Third column: 3D plot (x,y,t) from the two scenes.

**Figure 2 sensors-24-01382-f002:**
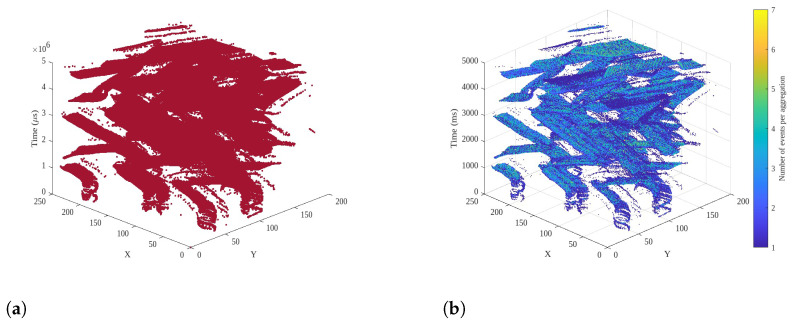
Point clouds associated with the *Shapes* scene (polarity 0). (**a**) No temporal aggregation of events; (**b**) temporal aggregation of events (20 ms).

**Figure 3 sensors-24-01382-f003:**
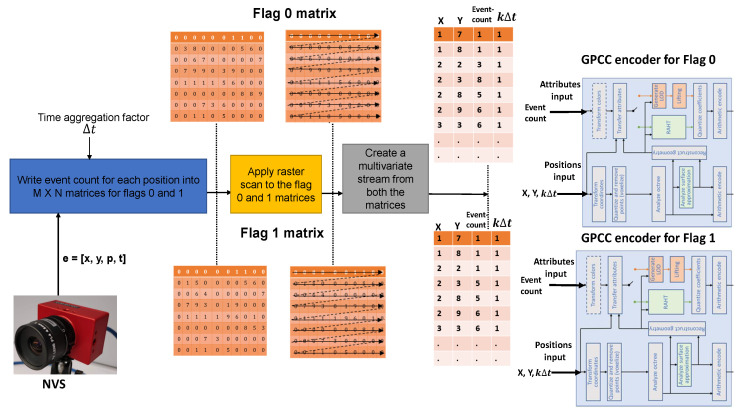
Proposed method (TALEN-PCC) of compression using 3D point cloud encoding [40] for time-aggregated NVS data. For a more detailed version of the G-PCC encoder section, see Figure 4.

**Figure 5 sensors-24-01382-f005:**
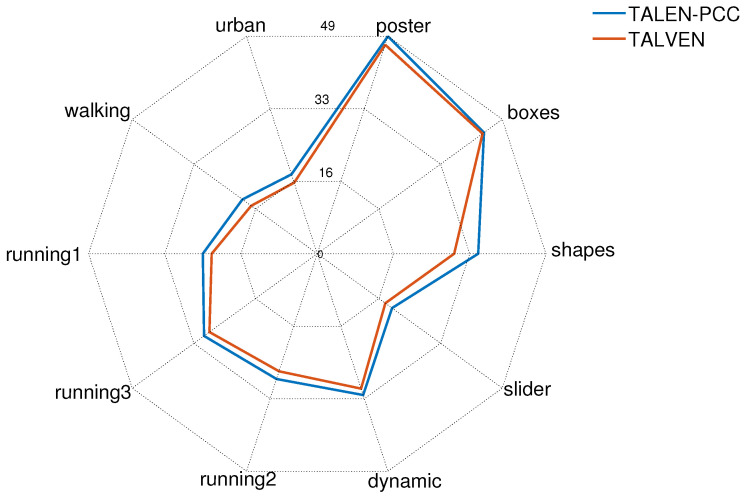
Compression ratio performance for the proposed strategy (TALEN-PCC) vs. TALVEN [17] for the ten scenes in Table 3. Aggregation time Δt=20 ms.

**Figure 6 sensors-24-01382-f006:**
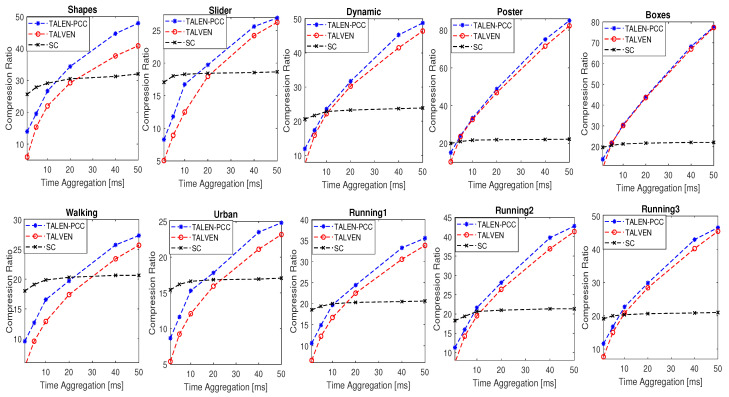
Compression ratio performance for the proposed strategy (TALEN-PCC), TALVEN [17], and Spike Coding (SC) [19] for different aggregation time intervals.

**Table 1 sensors-24-01382-t001:** Spike event stream sample.

*x*	*y*	*p*	*t*
5	18	1	45
7	17	0	48
1	10	0	50
2	20	0	55
8	20	1	56
1	10	0	58
8	11	1	61

**Table 2 sensors-24-01382-t002:** Algorithms employing the accumulation of spike events in diverse applications.

Paper	Algorithm	Application	Task	Spike Accumulation Interval (Δt)
[3]	Convolutional Neural Network (CNN)	Slip detection	Object vibration and stress distribution detection	10 ms
[9]	Synaptic Kernel Inverse Method (SKIM)	Visual classification	Digit classification	20 ms
[10]	Deep residual network (ResNet-50)	Autonomous driving	Motion estimation	50 ms
[11]	Time Delay Neural Network (TDNN)	Tactile sensing	Material classification and contact force estimation	7 ms
[12]	Asynchronous Convolutional Network (YOLE)	Object detection	Detection of objects, and prediction of their direction and position	10 ms
[16]	Long Short-Term Memory (LSTM) neural networks	Tactile sensing	Contact force estimation	10 ms

**Table 3 sensors-24-01382-t003:** Extracted dataset for experimental analysis. The indoor/outdoor sequences are ordered from higher to lower event rate.

	Sequence	Event Rate (kev/s)	Extracted Sequence Duration (s) and Start/End Time (s)	Scene Complexity	Speed
Indoor	Boxes (Rotation)	4288.65	5 (45–50)	High	High
	Poster (Rotation)	4021.1	5 (45–50)	High	High
	Dynamic (Rotation)	1077.73	20 (1–20)	Medium	Medium
	Slider (Depth)	336.78	3 (1–3)	Medium	Low
	Shapes (Rotation)	245.61	20 (1–20)	Low	Low
Outdoor	Running3	1525.5	20 (40–60)	Medium	High
	Running2	1229.4	20 (20–40)	Medium	Medium
	Running1	713.8	20 (1–20)	Medium	Medium
	Urban	503.04	10 (1–10)	High	Low
	Walking	342.2	20 (1–20)	Medium	Low

## Data Availability

Publicly available datasets were analyzed in this study. The data used in this paper can be found here https://rpg.ifi.uzh.ch/davis_data.html, accessed on 20 November 2023.

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
