# Peer review of "Lossless Encoding of Time-Aggregated Neuromorphic Vision Sensor Data Based on Point-Cloud Compression"

_sensors, 2024, doi:10.3390/s24051382_

Round 1

Reviewer 1 Report

Comments and Suggestions for Authors

Major revision

This manuscript introduces The proposed strategy Time-Aggregated Lossless Encoding of Events based on Point Cloud Compression (TALEN-PCC). In conclusion, the research is intriguing and yields valuable results; however, the current document has several weaknesses that need to be addressed in order to produce a manuscript that aligns with the significance of the findings.

General considerations:

(1) At a thematic level, the proposal offers a very interesting vision, as lossless compression of NVS data would be a very useful technique for engineering. However, lossless compression of NVS data is not only about the compression rate, but also about the speed and complexity of the compression algorithm. This issue is an important limitation about the aspirations of the proposal, whose limitations should be assumed with more rigour and realism in the development of the argumentation of the manuscript.

(2) In the "Contributions" section, please further describe the compression methodology used by TALVEN, in particular the specific steps of pulse event aggregation and video encoding. Provide more details or examples of how this method works.

Title, Abstract and Keywords:

(3) The title is clear and to the point. The abstract contains the purpose and conclusion of the study. Keywords fully cover the topic and core concepts of the paper. Nevertheless, the advantages and disadvantages of TALEN-PCC over other methods can be summarized in brief words in the abstract.

Chapter 1: Introduction

(4) The first paragraph introducing the research topic may present a much broad and comprehensive view of the problems related to your topic with citations to computer vision application references in various fields (A Performance Analysis of a Litchi Picking Robot System for Actively Removing Obstructions, Using an Artificial Intelligence Algorithm; Agronomy. Transforming unmanned pineapple picking with spatio-temporal convolutional neural networks. Computers and Electronics in Agriculture.).

(5)  In the section that introduces the NVS data representation and explains how it can be interpreted as a point cloud, two different scenarios (shape and dynamics) are mentioned, but there is not enough logical connection in the description to explain why these scenarios were chosen and how they demonstrate different aspects of the NVS data representation.

Chapter 3: Experiments and results

(6) At line 225, the process of converting 64-bit AER data to (x, y, p, t) multivariate streams should be described in sufficient detail and examples provided by the author. so that the process can be understood and repeated by others

(7) The authors mention the compression performance of different scenarios and the data features associated with the scenarios. However, there is not enough data support to demonstrate the relationship between compression performance and scene characteristics. There is no quantitative analysis or correlation corresponding to scene characteristics

Chapter 4: Conclusions

(8) Describe the performance of the technology in a real-world engineering environment, e.g., stability and reliability

(9) It should mention the scope for further research as well as the implications/application of the study.

(10) I recommend including the limitations regarding the consideration of damage indicated in this review in the limitations assessment. This part of the document can be improved and completed with more rigour.

Reviewer 2 Report

Comments and Suggestions for Authors

This manuscript proposes a lossless encoding method of time-aggregated Neuromorphic Vision Sensor (NVS) data based on point cloud compression. It mainly consists of three steps: First, turn the NVS data into flag zero and one matrices. Second, turn the matrices into multivariate streams. Third, feed the multivariate streams to G-PCC encoder in the lossless compression mode. The experiments show that the proposed method outperforms previous proposed methods TALVEN and Spike Coding. My concerns are: 1、Is it possible to combine the first two steps of proposed method with other encoders? 2、The G-PCC encoder parts in Figure 3 are too small to be clearly viewed. The figure need to be updated.

Reviewer 3 Report

Comments and Suggestions for Authors

===== Synopsis:

The authors improve one of their own methods for spike-event based compression and show the improved performance on a small set of indoor/outdoor scenes. The study reads fairly well.

===== General Comments:

I find the improvement rather minimal.

The study could be written a bit more carefully. Some acronyms are introduced without their abbreviation in text, though they appear in a table.

===== Specific Comments:

- Figure 1: I actually have troubles understanding it exactly, despite having seen the output of the Silicon Retina many times. For shapes it seems a small motion, for dynamic scene it appears much larger and seems to involve even a complete change in position (mirrored?).

Some explanations on how the motion occurs would be helpful. For example for shapes I have the impression the center of rotation is roughly the center point of the image plane.

- accumulation: it appears to me that it is a quantization, in particular if one uses a fixed time interval. Or would that term not be appropriate in this context.

- line 128: "very high" file size. Say huge.

- lines 161: better display as table, it's hard to read from a single line.

Comments on the Quality of English Language

no particular comments.

Round 2

Reviewer 1 Report

Comments and Suggestions for Authors

accept